# Heart and Lung Sound Measurement Using an Esophageal Stethoscope with Adaptive Noise Cancellation

**DOI:** 10.3390/s21206757

**Published:** 2021-10-12

**Authors:** Nourelhuda Mohamed, Hyun-Seok Kim, Kyu-Min Kang, Manal Mohamed, Sung-Hoon Kim, Jae Gwan Kim

**Affiliations:** 1Biomedical Science and Engineering Department, Gwangju Institute of Science and Technology, Gwangju 61005, Korea; nonoalhodaali@gm.gist.ac.kr (N.M.); manalalnosh@gm.gist.ac.kr (M.M.); 2Biomedical Engineering Research Center, Asan Institute for Life Science, Asan Medical Center, Seoul 05505, Korea; hskim-bme@amc.seoul.kr; 3Department of Anesthesiology and Pain Medicine, University of Ulsan College of Medicine, Asan Medical Center, Seoul 05505, Korea; km.kang@amc.seoul.kr

**Keywords:** digital esophageal stethoscope, esophageal catheter, adaptive noise canceling, least mean square, SIMULINK

## Abstract

In surgeries where general anesthesia is required, the auscultation of heart and lung sounds is essential to provide information on the patient’s cardiorespiratory system. Heart and lung sounds can be recorded using an esophageal stethoscope; however, there is huge background noise when this device is used in an operating room. In this study, a digital esophageal stethoscope system was designed. A 3D-printed case filled with Polydimethylsiloxane material was designed to hold two electret-type microphones. One of the microphones was placed inside the printed case to collect the heart and lung sound signals coming out from the patient through the esophageal catheter, the other was mounted on the surface of the case to collect the operating room sounds. A developed adaptive noise canceling algorithm was implemented to remove the operating room noise corrupted with the main heart and lung sound signals and the output signal was displayed on software application developed especially for this study. Using the designed case, the noise level of the signal was reduced to some extent, and by adding the adaptive filter, further noise reduction was achieved. The designed system is lightweight and can provide noise-free heart and lung sound signals.

## 1. Introduction

A patient under general anesthesia is monitored by using a variety of techniques to observe the effect of the anesthetic and to safeguard the patient against progression into adverse conditions [1]. Auscultating heart sound and lung sound using an esophageal stethoscope, which is medical equipment for auscultation or listening to internal heart and lung sounds, is a simple, inexpensive, and much more noise-free method to provide information about cardiac sounds, but it requires highly experienced and practiced anesthetists to obtain accurate results [2]. Thus, there have been many studies on developing digital esophageal stethoscope systems that can accurately measure heart and lung sound signals. In previous studies, it has been shown that digital esophageal stethoscope systems can help to detect the intratracheal accumulation of secretion [3], to measure cardiovascular function [4,5], and predict fluid responsiveness during surgery [6]. However, when using a digital esophageal stethoscope in an operating room, there is a lot of noise coming from outside of the patient, which corrupts the main heart and lung sound signals. Adding sound-absorbing materials and some filtering techniques can be effective in reducing the noise. Porous materials, which have sound absorption properties, can be considered as one of the most effective methods for noise reduction [7,8] because they reduce the energy of the noise signal and gradually weaken it as the thickness of the material increases [9]. To choose one of the available filtering techniques, the frequency ranges of the heart and lung sound signals need to be considered. The heart sound frequency range is generally 5–600 Hz [10], and the lung sound frequency for healthy subjects can extend up to 1000 Hz; in abnormal cases, sounds can appear at frequencies above 2000 Hz [11]. One of the available filtering techniques is to apply an analog band-pass filter. Although an analog band-pass filter can remove a relevant band of interference, it also has an impact on the main signal which overlaps in the filtered frequency band. For example, the commonly used band-pass filter for heart sound measurements allows frequencies between 30 and 200 Hz to pass while rejecting the others [12]. If this filter is used, it can remove some important information (see Figure 1); additionally, the filtered band can still have noisy contents [12]. The adaptive noise canceling algorithm, on the other hand, is one of the most popular techniques used for the effective removal of noise corrupted with the main signal [13]. In the proceeding sub-sections, an introduction to the basic ideas and techniques used in this study is provided.

### 1.1. Heart and Lung Sound Recording Techniques

Heart sound, which is made by the opening and closing of the heart valves during the diastole-systole phases of the cardiac cycle and the blood flow [14], and lung sound, which is created by breathing, can be auscultated by using chest stethoscopes, esophageal stethoscopes, and phonocardiography.

There are different types of chest stethoscopes, such as ordinary acoustical stethoscopes and digital stethoscopes. The ordinary acoustical stethoscope is a simple device that conveys the energy of the sound created in the patient’s chest to the ear of the physician using an air column. In order to conduct good measurements, a highly experienced physician is required, and this makes the process highly subjective. Additionally, the signal is prone to be corrupted with noise from outside, low-frequency sounds cannot be detected by the physician, and the measurement cannot be recorded for further analysis. For digital stethoscopes, although the signal can be recorded for future use, it is still prone to outside noise because the measurements are performed from the chest. An esophageal stethoscope that acquires the heart and lung sounds from the esophagus, which is very close to the heart, can provide a signal which is less affected by the operating room noise. In one configuration, the esophageal catheter is directly connected to the stethoscope. However, this also requires highly experienced physicians in order to obtain good results, and still cannot provide recorded data to be used in the future.

Digital esophageal stethoscopes, in which the heart and lung sound signals are recorded using the esophageal catheter, processed, and then displayed using digital means, do not depend directly on the expertise of the physician to obtain accurate measurements, can detect even low-frequency heart sounds, and can also provide records that can be used for further analysis.

### 1.2. Sound-Absorbing Materials

Collecting heart and lung sound signals from a patient in an operating room is so challenging because the sound signal is prone to be corrupted with other operating room sound signals coming from different sources, such as other monitoring equipment sounds, physician and nurse speech, operating room-warmer sounds, to name a few [15]. Many contemporary studies have focused on the elimination of vibrations and noise [16] by different means, such as using sound-absorbing materials. Sound-absorbing characteristics of materials are generally affected by many factors, including the material thickness, material density, and material porosity [9], which is why materials containing pores, such as polyurethane and polymer foams, are commonly studied to prove their ability to absorb sound. Generally, it was found that these materials at low frequencies exhibit poor acoustic performance [16,17]. Polydimethylsiloxane (PDMS) material is a two-component (part A and part B) silicone foam that cures at room temperature by an addition cure reaction. Generally, the two components are mixed in a ratio of 1:1 to form a foamed elastomer. The authors in [16] investigated the acoustic behavior of PDMS material, mixing part A, part B, and PDMS oil in various ratios and preparing different samples with different names (provided in Table 1). The sound absorption coefficients of these samples were measured and are provided in Figure 2. They found that reducing the ratio of the part A component of the PDMS material can improve the sound absorption coefficient in low frequencies, whereas adding a thinning agent, such as PDMS oil, in a ratio of 5% to 25%, can improve the material sound absorption property for high frequencies [16].

### 1.3. Adaptive Noise Cancellation

One of the most popular applications of adaptive filters is noise cancellation. The main goal of adaptive noise cancellation is to find the best estimate for noise corrupted with the main signal and then subtract this estimated noise from the desired signal in order to obtain a noise-free signal [13]. To do so, the filter implements one of the adaptive algorithms to vary the values of the filter coefficients until a better approximation of the signal is maintained after a certain time of iterations [18]. In general, these filters have two parts: a digital filter and an adaptive algorithm, as shown in Figure 3. In this diagram, *d*(*n*) represents the main signal, which is the desired signal corrupted with noise, *x*(*n*) is the noise signal, *y*(*n*) is the estimated noise signal after applying the adaptive filtering algorithm, and the *e*(*n*) is the reference signal. The two input signals, $d\left(n\right)$ and $x\left(n\right)$*,* and $e\left(n\right)$, which is used to adjust the coefficients of the digital filter [19], are required to implement these filters. The reference signal takes different values in each iteration until the specified number of iterations is reached; at this point, the best estimate of the noise signal is reached, and the final filter output can be obtained. Sometimes, the term “reference signal” is given to the noise signal *x*(*n*) [20].

There are two main categories of adaptive filtering algorithms: least mean square (LMS) and recursive least squares (RLS). Each algorithm has its own pros and cons. LMS algorithms, on the one hand, are computationally simple and easy to understand and implement, whereas RLS algorithms are more complex, computationally expensive, and require huge memory [20].

LMS is an approximation of the steepest descent algorithm, which uses an instantaneous estimate of the gradient vector [13]. To perform the estimation, an input sample signal and an error signal are required. The algorithm changes the filter coefficient many times, in iterations, moving it toward the approximated gradient. The main goal of the algorithm is to minimize the cost function. To update the filter coefficients values in the LMS algorithm, Equation (1) is used, in which $\vec{w}\left(n+1\right)$ is the next coefficient vector, $\vec{w}\left(n\right)$ is the current coefficient vector, µ is the step size, $x\left(n\right)$ is the input signal, and $e\left(n\right)$ is the error signal. Before using this equation, two important parameters need to be determined: the filter length (L), which controls the number of iterations, and µ, which determines the convergence speed of the algorithm [21], and is a small positive number that exactly controls the influence of the filter coefficient updating factor. The selection of an appropriate value for µ directly affects the performance of the LMS algorithm; if this value is too small, the adaptive filter will take a long time to converge toward the optimal solution, and the filter becomes unstable and has diverged output if the value is too large [20]. To determine the speed of the LMS algorithm and better understand the performance of the filter, the learning curve is used, which is a plot of the mean square error (MSE) of the adaptive filter versus time or iteration, and this curve decays exponentially to a constant value.
(1)$$\vec{w}\left(n+1\right)=\vec{w}\left(n\right)+\mu x\left(n\right)e\left(n\right)$$


Thus, in this project a digital esophageal stethoscope system was designed that can record heart and lung sound signals by means of microphones, which were held inside a printed case filled with PDMS material, and a processing unit that performed the adaptive noise canceling algorithm with the output signal displayed using software developed especially for this purpose.

## 2. Materials and Methods

This study aimed to design a digital esophageal stethoscope system in order to record heart and lung sounds. To build this system, an esophageal catheter, two electret condenser-type microphones, a processing unit, and a developed software application were used. The esophageal catheter was used to measure the heart sound. For the two microphones, the first was connected, inside a 3D-printed case and filled with PDMS material, to the end of the esophageal catheter to collect cardiopulmonary sound signals, while the other microphone was mounted on the surface of the case to pick up the other operating room sounds. The signals picked up from the two microphones were then fed to the processing unit, which mainly consisted of an amplification circuit to amplify the recorded signals to a certain level, and an adaptive noise canceling algorithm code was implemented using the Teensy 3.2 development board to filter out the operating room noise from the main heart and lung sound signals. The output of the system was then displayed using software developed for this study. The displayed data include raw heart sound signal, noise signal, noisy heart sound signal, and the recovered (noise-free) signal after applying the adaptive filter. Figure 4 shows the basic concept of the system.

### 2.1. Microphone Case

The aim of designing this case was to reduce the noise corrupted by the main heart and lung sound signals and to hold the two microphones that collected the signals which were required to implement the adaptive filter. The outer layer of the microphone case was designed using Fusion 360 modeling software and printed using TPU 3D printing material; then, it was filled with PDMS material to absorb the noise. To form the filling mixture, 50% of part A, 50% of part B of PDMS material, and 10% of PDMS oil were mixed together. This mixture was prepared according to the study in [16], discussed earlier in the Introduction, and its sound absorption coefficient in the frequency range of interest is relatively high. For the two microphones, the first microphone was placed inside the microphone case in order to collect the heart and lung sound signals coming through the esophageal catheter, although this microphone could still collect noise coming from the outside environment; thus, its signal is the main adaptive filter signal. The second microphone was mounted on the surface of the case close to the first microphone so that it could capture the same noise picked up by the first microphone; this microphone would probably only pick up the sound coming from the environment, which is the noise signal. Figure 5 shows the designed case drawing and the fabricated model. For better fixing the esophageal catheter inside the case, a fixing part was added.

After designing the microphone case, its resonance frequency was measured in the frequency range of interest using Laser Doppler Vibrometer and the acoustic excitation method. It was found that the resonance peak was at a frequency of 57 Hz, as shown in Figure 6.

### 2.2. Microphone

Analog electret condenser microphones have a wide range of packaging, directionality options, and a relatively long lifespan, which is why they are preferred in many applications. In this study, a POM-2738L-LW100-R analog electret condenser microphone was used. It was chosen because of its appropriate frequency response range, diameter, weight, and price. The used microphones’ frequency characteristic curve and specifications are provided in Figure 7 and Table 2, respectively. It is clear from the specification and curve that it could accurately pick up the heart and lung sound signals. The diameter of the microphone is also important. The inner diameter of the used esophageal catheter was 4 mm; therefore, a microphone that has a similar or close diameter is important in order to reduce the noise coming from the outside of the catheter. The used microphone had a diameter of 6 mm; of course, it would have been better to use a 4 mm diameter microphone, but the available ones on the market have a higher frequency range, so if used, some of the heart sound signals may have been unpicked. Additionally, the used microphone was lightweight and low cost, which better served the purpose of this study.

### 2.3. Processing Unit

The recorded signals from the two microphones were fed into the processing unit. The first step of the processing unit is the amplification circuit, which is a power amplifier used to amplify the microphones’ sensed signals to a level that can be detected by the microcontroller. The schematic diagram of the circuit is provided in Figure 8. An LM324 operational amplifier was used because it has a high frequency response in the frequency range from 1 Hz up to 10 kHz (see Figure 9a), which covered the frequency range of interest of this study. Figure 9b shows the overall power amplifier circuit frequency characteristic graph which was made using Proteus 8 professional software after simulating the circuit; it also shows a flat frequency response in the frequency range from 10 Hz up to 10 kHz which covered the frequency range of interest of this study.

The second step of the processing unit involved the filter. In this study, an adaptive noise canceller based on the least mean square algorithm (ANC-LMS) was implemented. There are many options for the CPU boards that can be used to implement ANC-LMS filters, such as Arduino, STM32F103×8 digital signal processors, and Teensy development boards. For this study, the most important issues were the board processing speed, size, and weight. The Teensy 3.2 was used because it has:

1. High processing speed:

The Teensy 3.2 has a processing speed of 72 MHz, which is a very high processing speed that allows real-time display for the output.

2. Small size and lightweight.

The proposed system was directly connecting the microphone case to the processing unit and then connected to the PC through the USB; therefore, this part of the system (microphone case + processing unit) needed to be as small and lightweight as possible. Additionally, because it was connected to the esophageal catheter, which was inserted into the patient’s esophagus, there must be no chance for it to detach from the catheter during the measurement. Although some other Arduino boards, such as DFR0282 Beetle, is smaller, the existence of the amplifier circuit combined with the CPU, which cannot be very small, needs to be considered.

3. Real analog output pin (A14):

Using this pin, an analog output sound from the system could be obtained, which necessitated developing other software to display the output signal in the future.

To write the software code of the ANC-LMS filter, the optimal values for the filter length (L) and the step size (µ) needed to be determined. The authors in [13], in order to find the appropriate L and µ values, developed a SIMULINK model to simulate the data and test the algorithms. They used simulated heart sound data with a length of 10 s and four types of external noises (white, pink, babble, and factory). They varied the step size value from 0.001 to 1 with a constant filter length of 32; then, the appropriate step size was determined by the signal-to-noise ratio (SNR). According to their findings, the appropriate step size and filter length for the LMS algorithm was found to be 0.01 and 32, respectively. Similarly, a two-stage SIMULINK model was built [23] in this study using MATLAB R2021a. The first stage block of the model was made to provide simulated data to the system, whereas the second stage block was made to test the LMS algorithm. Figure 10 shows a block diagram of the implemented model. Additionally, different values for µ and L were tested, and similarly, the best step size and filter length values were found to be 0.01 and 32, respectively. After that, the software code was written in C programming language and uploaded to Teensy 3.2 using Arduino IDE software, version 1.8.13.

In the software code, the sample window used to display the output of the filter was 50 ms. Additionally, the speed of the LMS algorithm was measured using the demo provided in the LabVIEW 2021 software documentation, and the learning curve was obtained and shown in Figure 11. This figure shows that the used LMS algorithm will give the same results after about 800 iterations.

Finally, a box printed with the 3D printer was used to hold the two main parts of the processing unit and attach them to the microphone case.

### 2.4. System Software

The output signal of the processing unit was displayed in a developed MATLAB app designed using the MATLAB R2021a software environment. This app was designed using App Designer, which is an interactive development environment for designing app layouts and programming its behavior and some commands with built-in MATLAB functions, such as serial function, evalin function, plot function, and strcmp function. The app could display the signal in real time, and using it, the user could enter the date, write the file name in which the recorded signal could be saved, choose the port number (COM number) from which the signal was measured, connect to the serial port, start the record and save the data.

## 3. Results

### 3.1. System Hardware

Figure 12 shows the different parts implemented and used in the digital esophageal stethoscope system. Figure 12a shows the fabricated microphone case with the added fixing part while holding the esophageal catheter, Figure 12b shows the implemented amplification circuit, and Figure 12c shows the Teensy 3.2 development board. Figure 13 shows the assembled parts of the overall system.

After establishing the system, a specific setup was used to collect and display the results. Figure 14 shows a block diagram of the used system setup. As the figure implies, two sound sources were used to play real sounds recorded in the operating room for a length of 7 min. Specifically, the first source simulated a patient’s heart sound by playing two normal heart sounds, S1 and S2, recorded using the esophageal stethoscope at a frequency of 4000 Hz. S1 was caused by the closure of the mitral and tricuspid valves, and S2 was caused by the closure of the aortic and pulmonic valves [24]. The second source simulated noise by playing recorded sounds that came from both staff and equipment. The major component of staff-related noise was conversations, whereas equipment-related noise was caused by air-warmers and the patient monitors’ alarm [15].

Using MATLAB, the parameters of the used signals, from the two sources, and their spectral densities were obtained and shown in Figure 15. The first signal had an amplitude of 1 Vpp and a main frequency of around 82 Hz (see Figure 15a), and the second signal had an amplitude of 1 V, and a main frequency of around 222 Hz (see Figure 15b).

### 3.2. PDMS Microphone Case Results

In order to verify that adding the PDMS material to the microphone case will reduce the noise by absorbing the external sounds, an experiment was performed using the case before and after filling it with PDMS material. A graph showing the relationship between the sound level in dB and frequency in Hz was obtained and is shown in Figure 16. Short-time Fourier transform (STFT) was used to compute the spectrogram of this figure, the number of samples per second was 4000 samples and the FFT size was 2048 samples, so the FFT was calculated every 2048/4000 = 0.512 s. The used frequency resolution was 4000/2048 = 1.953125 Hz.

### 3.3. SIMULINK Model Results

By using the two-block model stated in the Materials and Methods section, and by setting the values of the step size and filter length, the model results were obtained and are shown in Figure 17.

### 3.4. System Experimental Results

In this part, the recovered signal from the developed system will be shown. To do so, all system parts were assembled as in Figure 13, the system setup was established as in Figure 14, the code was uploaded in the Teensy 3.2, and the developed MATLAB app was used to display the recorded system output. In the MATLAB app, the date was set, the file name and COM port number were provided and the CONNECT button was pressed to connect to the specified port, and then the START button was pressed to start the measurement. Sounds were played only for 10 s. After the measurement was finished, the system was disconnected, and the record was saved using the SAVE button.

Figure 18 shows the experimental system results. It contains four main graphs: the first graph shows the original signal which was measured while only playing the first sound source, and it represents the desired heart sound signal. The second graph shows the noise signal, which was measured from the second sound source only, and it represents the background noise, i.e., the operating room noise. The third graph shows the sum of the main signal coming from the first sound source and the noise signal coming from the second sound source, and it was measured while the two sound sources were on. The fourth graph shows the adaptive filter-recovered output signal, which was measured while the two sources were on, and it can be seen that the noise gradually decreased and the signal was almost similar to the desired signal presented in the first graph.

## 4. Discussion

Auscultation of heart and lung sound is a very popular technique established a long time ago to diagnose heart valve-related problems; however, it requires highly experienced, practiced, and professional physicians to gain good diagnosis results [25]; moreover, the acquired signals cannot be saved for future use. The current trend of electronic stethoscope system development is a major advancement from traditional acoustic stethoscopes because they allow hearing, viewing, recording, and transfer of the heart and lung sound signals. Additionally, further analysis and feature extraction algorithms can be applied to the data for diagnosis purposes [26]. This project aimed to design a digital esophageal stethoscope system that can acquire heart and lung sound signals from the patient’s esophagus, which is close to the heart, filter them, and display the results in the developed software. The implemented filtering technique is a popular technique used for audio signal processing to remove noise. It is an advancement over other techniques because it better estimates the noise signal and subtracts it from the main signal which is corrupted with noise to gain a noise-free signal. Using MATLAB 2021a software, the signal-to-noise ratio (SNR) was calculated before and after applying the filter in the SIMULINK model. The power of the noisy input signal, the filter output signal and the noise signal was calculated using fft and fftshift built-in MATLAB functions. Then, using the fact that SNR is equal to signal power divided by the noise power, the SNR was calculated. The SNR value before filtering was found to be 1.0992 dB, whereas the SNR value after filtering was 1.9745 dB, which is much better than without filtering. Using this system, the patient’s noise-free heart and lung sound signals can be recorded, displayed in real-time, and saved for future use.

## 5. Conclusions

In this project, a lightweight digital esophageal stethoscope system with an adaptive noise cancellation was designed. Using this system, heart and lung sound signals could be recorded in real time and also stored for future use. The subject’s heart and lung sounds were collected using a microphone connected to the end of an esophageal catheter, which was placed inside the esophagus near the heart. Another microphone was used to pick up the operating room sounds that can corrupt the main signal. The output of the two microphones was then fed to the processing unit to be amplified and filtered. Finally, the output was displayed using the software application developed for this project, which can display real-time waveforms for the original signal, noise, noisy signal, and filtered signal.

The results showed that the developed adaptive filter for noise-canceling worked properly. Most of the noise corrupted with the main signal was removed after using the PDMS material as a sound-absorbing material and applying the adaptive filter.

In the future, we will optimize the quality of the recovered signal and obtain clinical data using the system to better verify the system results.

## Figures and Tables

**Figure 1 sensors-21-06757-f001:**
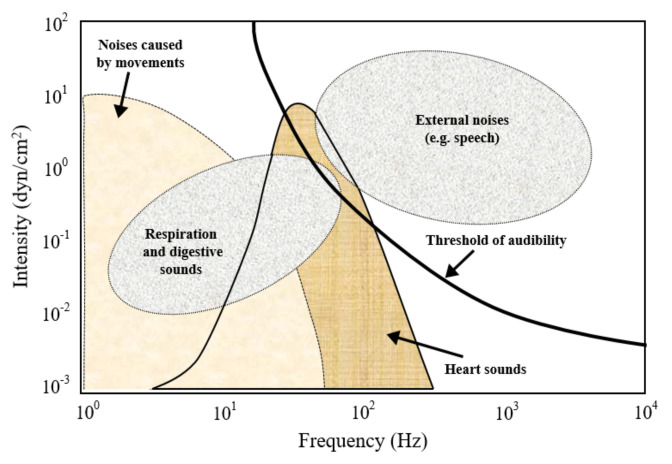
Spectral intensity map of phonocardiographic records [12].

**Figure 2 sensors-21-06757-f002:**
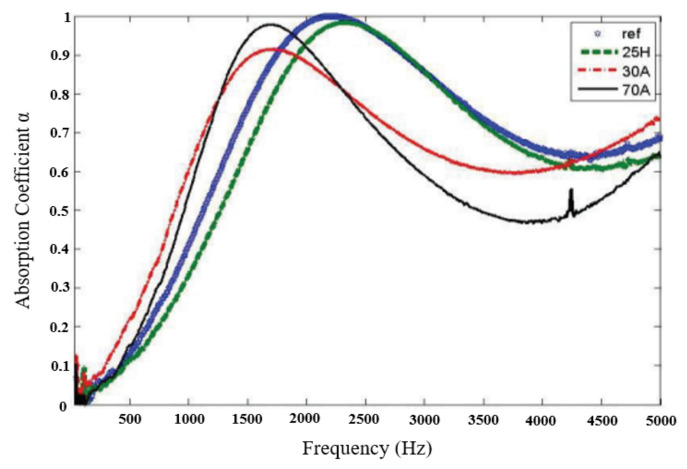
Sound absorption coefficient of the different samples according to [16].

**Figure 3 sensors-21-06757-f003:**
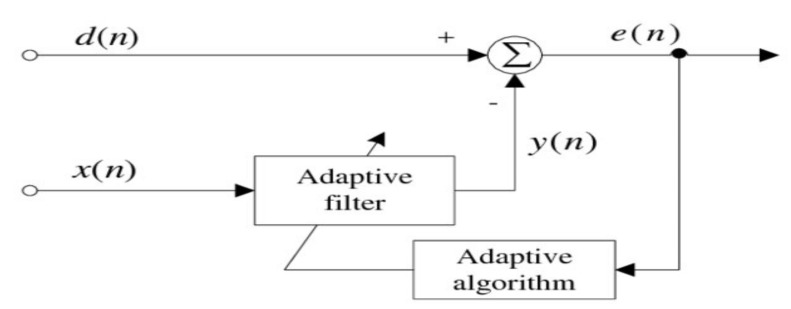
General block diagram of an adaptive filter.

**Figure 4 sensors-21-06757-f004:**
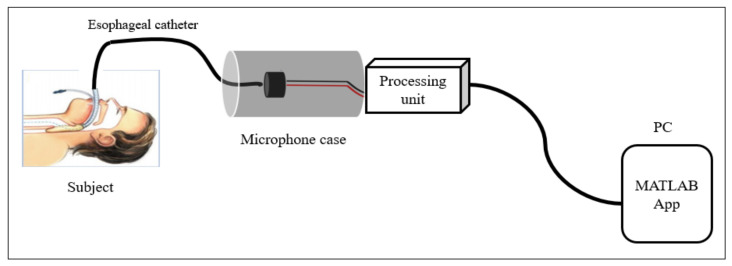
Basic concept of the digital esophageal stethoscope system.

**Figure 5 sensors-21-06757-f005:**
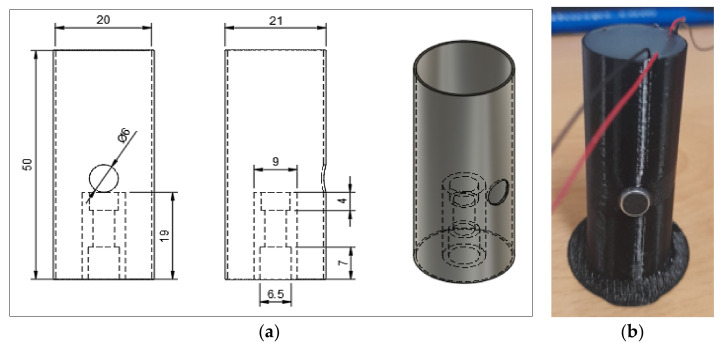
Microphone case drawing (**a**,**b**) case filled with PDMS.

**Figure 6 sensors-21-06757-f006:**
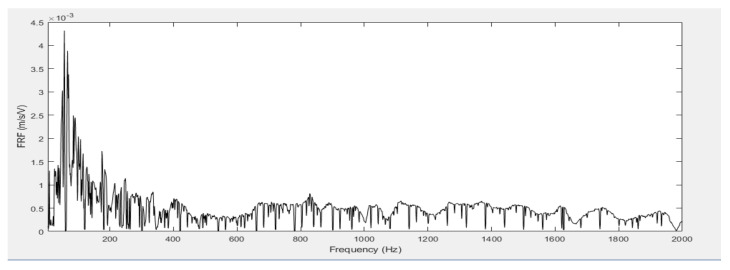
Resonance frequency of the casing.

**Figure 7 sensors-21-06757-f007:**
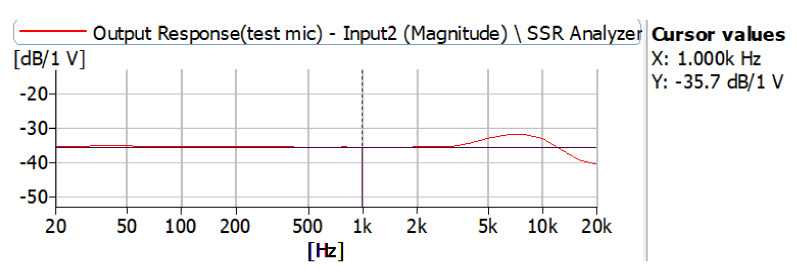
Used microphone frequency characteristic curve.

**Figure 8 sensors-21-06757-f008:**
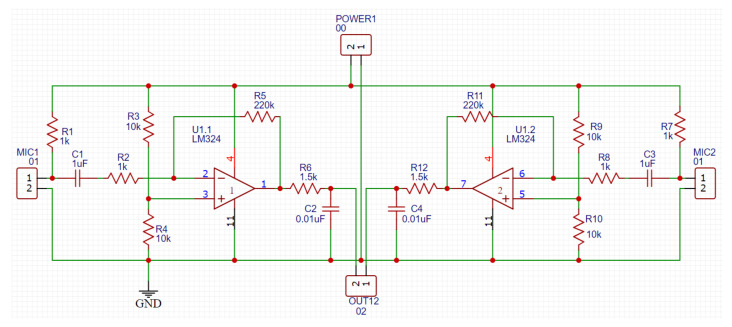
Schematic diagram of the power amplifier circuit.

**Figure 9 sensors-21-06757-f009:**
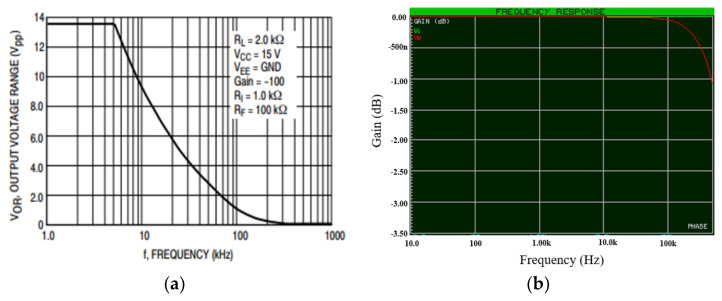
(**a**) Amplifier IC frequency response curve [22] and (**b**) power amplifier frequency response curve.

**Figure 10 sensors-21-06757-f010:**
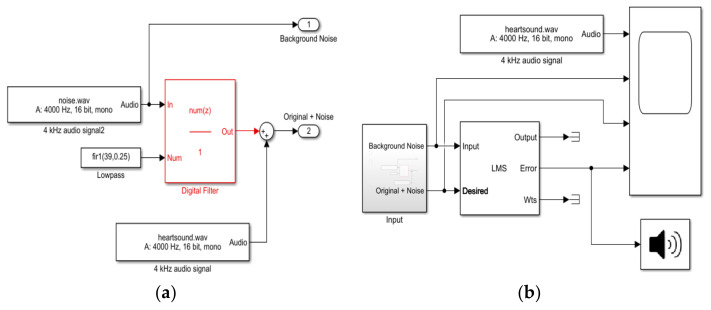
SIMULINK model: (**a**) block model designed to generate the simulated data and (**b**) LMS testing model.

**Figure 11 sensors-21-06757-f011:**
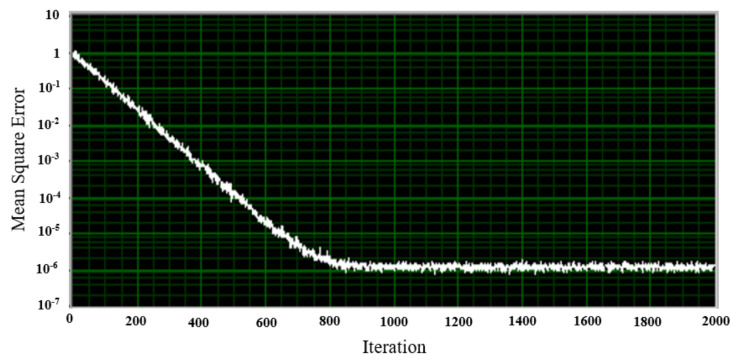
Learning curve of the LMS adaptive filter used.

**Figure 12 sensors-21-06757-f012:**
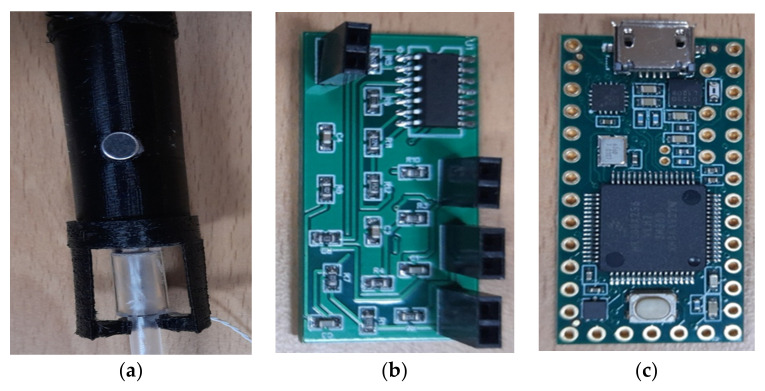
Different parts of the designed system. (**a**) Microphone case, (**b**) amplification circuit and (**c**) Teensy 3.2 development board.

**Figure 13 sensors-21-06757-f013:**
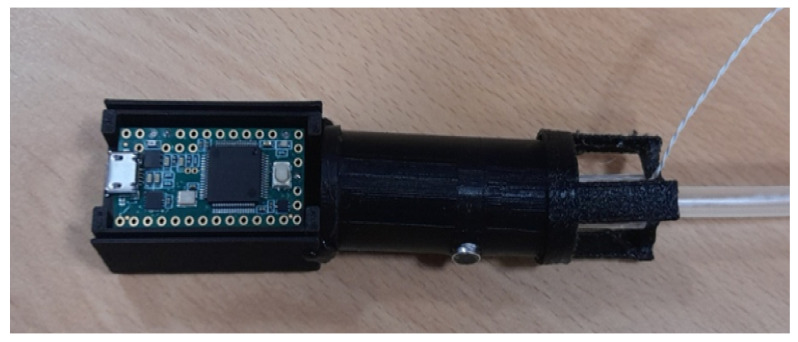
Developed digital esophageal stethoscope system hardware.

**Figure 14 sensors-21-06757-f014:**
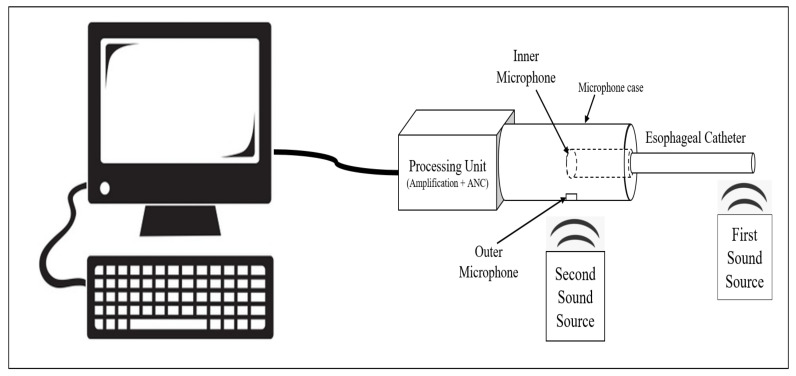
System setup used while collecting and displaying the results.

**Figure 15 sensors-21-06757-f015:**
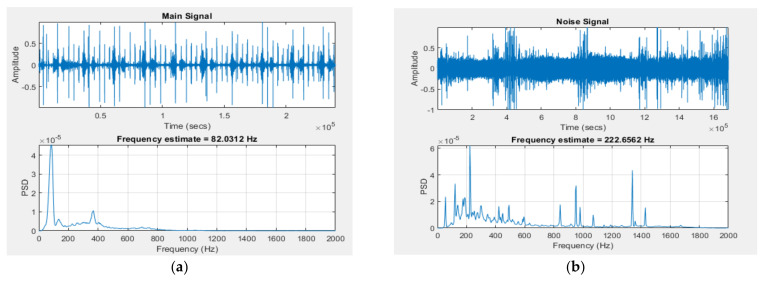
(**a**) Main signal and its power spectral density; (**b**) noise signal and its power spectral density.

**Figure 16 sensors-21-06757-f016:**
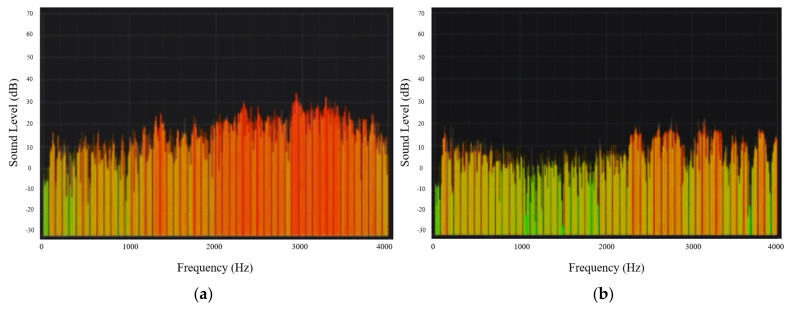
A graph showing sound frequency vs. sound level measured using the case (**a**) without PDMS material and (**b**) with PDMS material.

**Figure 17 sensors-21-06757-f017:**
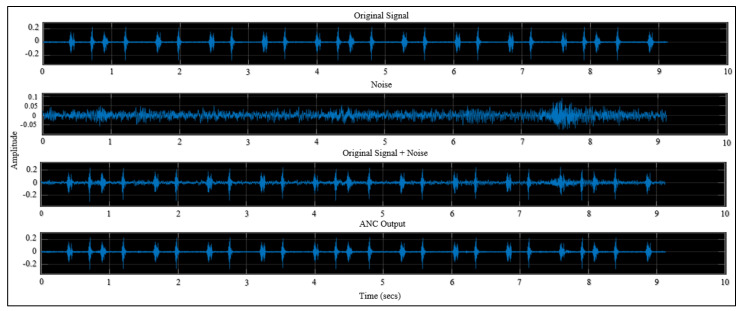
SIMULINK model result.

**Figure 18 sensors-21-06757-f018:**
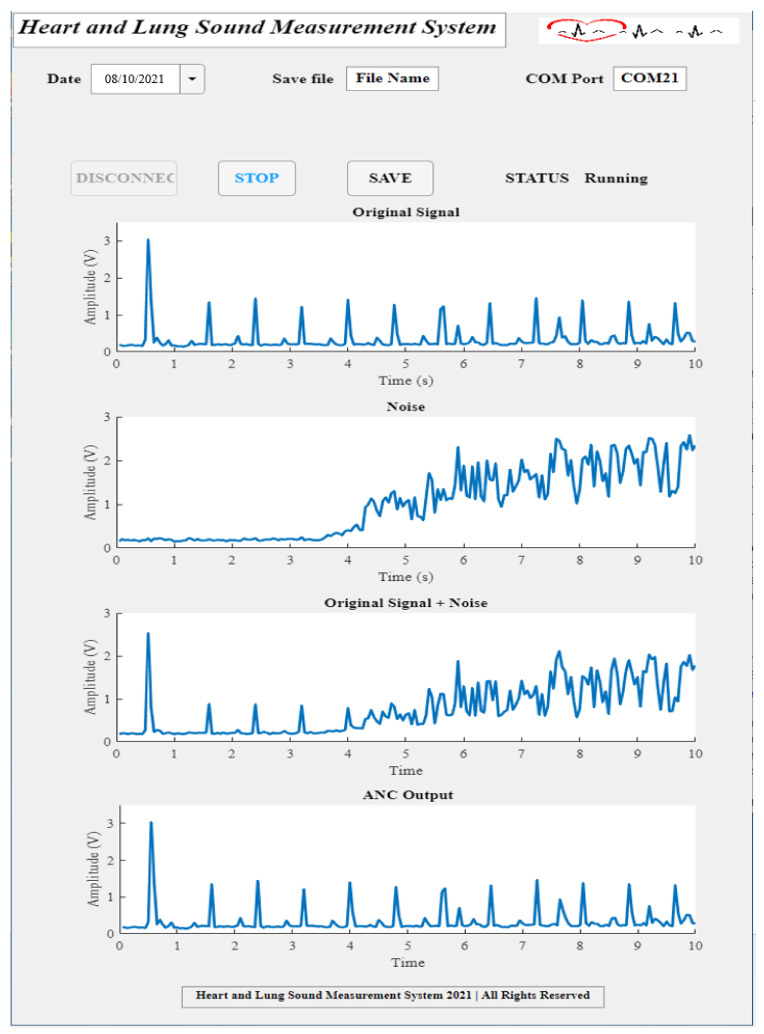
System experimental results displayed using the developed MATLAB app.

**Table 1 sensors-21-06757-t001:** Different samples used in the reference study [16].

Sample	% A	% B	Oil (% in Total Mass A + B)
ref	50	50	-
30A	30	70	-
70A	70	30	-
5H	50	50	5
25H	50	50	25

**Table 2 sensors-21-06757-t002:** Microphone specifications.

Parameter	Value	Unit
Directivity	Omni	-
Sensitivity	−38 ± 3 (12.6 mV/Pa)	dB
Standard operating voltage	2	Vdc
Max operating voltage	10	Vdc
Impedance	2.2	KΩ
Signal-to-noise ratio	>60	dB
Frequency response	15–16,000	Hz

## Data Availability

The data presented in this study are available on request from the corresponding author. The data are not publicly available because it is still under collection.

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
