# Peer review of "Heart and Lung Sound Measurement Using an Esophageal Stethoscope with Adaptive Noise Cancellation"

_sensors, 2021, doi:10.3390/s21206757_

Round 1

Reviewer 1 Report

Nice and inspirative paper. I have just one question what I did not find the answer in your text for: how is the reference signal obtained? Please, add a short explanation, because it is obviously crutial both for proper function of the algorithm and final interpretation of obtained oucomes. Thank you.

Reviewer 2 Report

The resonance frequency of casing and absorption coefficient in frequency range of interest of material used in casing should be added to the paper. Frequency characteristic of microphones and processing unit (amplifier) should be added to the paper.(modelled or measured). The frequency resolution and used FFT algorithm in figure 9 is not shown, (which averaging is used, time window). Explain better the usage of two microphones at two different places (the background noise level is much higher on the second microphone than on the first or). The sensitivity of microphones should be given in mv/Pa. Why analog electret microphones are used? How the SNR has been found in figure 11., it is visible some improvement but some parameters of noise signals and measured signal should be added in paper, the results in Figure 11 should be explained better. What is frequency content of the background noise (is this some fan or)? The paper seems interesting regarding implementation of ANC algorithm in practice but some additional things should be added, speed of algorithm, should be mentioned, why this DSP Teensy board has been used for implementation?

Reviewer 3 Report

Main message of the article

This article presents the design a novel digital esophageal stethoscope. 

General Judgment Comments

Overall, the article is overall good and well written, although more references may be required to give more strength to the authors’ assumptions and claims.

Suggestion: minor revision 

The article is overall of sufficient quality. I recommend the editor to accept the manuscript given that some revisions are made.

Major Issues

  • Some claims are under-referenced. For example, in Line 45, is there any paper reporting the effectiveness of sound-absorbing materials? Similarly on Line 48, are there clear examples of noise content in filtered bands? Authors should consider expanding the list of referenced materials to give more strengths to their claims (other examples on Line 82, 86, 99, … ). 
  • For clarity, authors should consider adding a brief summary of the findings of the referenced work in Line 191.

Minor Issues

  • On line 87, I believe the correct shortened version of Polydimethylsiloxane is PDSM, as also indicated in the following lines of the paper (e.g. Line 90). 
  • Line 109, I believe the “;” should have been “:”
  • Line 169, Brackets are not required after Table. 
  • Figure 5A and 5B, the word Audio covers the word Mono. 
  • In figure 9, the color choice makes it difficult to clearly see the spectrum. Authors should consider either changing the color of the plot or make the figure brighter.    
  • authors should consider indicating the version of the software and libraries used to enhance the reproducibility of their work (e.g. Matlab, App designer, etc).

Final comments

The article is overall good and presents an interesting project. Minor edits are suggested before the article is accepted for publication.

Round 2

Reviewer 2 Report

Everything is answered in response letter.